# Metabolomic Analysis Reveals the Metabolic Diversity of Wild and Cultivated Stellaria Radix (*Stellaria dichotoma* L. var. *lanceolata* Bge.)

**DOI:** 10.3390/plants12040775

**Published:** 2023-02-09

**Authors:** Zhenkai Li, Hong Wang, Lu Feng, Haishan Li, Yanqing Li, Gege Tian, Pilian Niu, Yan Yang, Li Peng

**Affiliations:** 1School of Life Sciences, Ningxia University, Yinchuan 750021, China; 2Ningxia Natural Medicine Engineering Technology Research Center, Yinchuan 750021, China

**Keywords:** medicinal plants, *Stellaria dichotoma* L. var. *lanceolata* Bge., metabolomic, metabolic diversity, wild, cultivation

## Abstract

Stellaria Radix, called Yinchaihu in Chinese, is a traditional Chinese medicine, which is obtained from the dried roots of *Stellaria dichotoma* L. var. *lanceolata* Bge. Cultivated yinchaihu (YCH) has become a main source of production to alleviate the shortage of wild plant resources, but it is not clear whether the metabolites of YCH change with the mode of production. In this study, the contents of methanol extracts, total sterols and total flavonoids in wild and cultivated YCH are compared. The metabolites were analyzed by ultra-high performance liquid chromatography–tandem time-of-flight mass spectrometry. The content of methanol extracts of the wild and cultivated YCH all exceeded the standard content of the Chinese Pharmacopoeia. However, the contents of total sterols and total flavonoids in the wild YCH were significantly higher than those in the cultivated YCH. In total, 1586 metabolites were identified by mass spectrometry, and 97 were significantly different between the wild and cultivated sources, including β-sitosterol, quercetin derivatives as well as many newly discovered potential active components, such as trigonelline, arctiin and loganic acid. The results confirm that there is a rich diversity of metabolites in the wild and cultivated YCH, and provide a useful theoretical guidance for the evaluation of quality in the production of YCH.

## 1. Introduction

Yinchaihu (Radix Stellariae) is a root medicine commonly used in Chinese traditional medicine. Traditional uses include treating fever and malnutrition, and it has been found to have anti-inflammatory, anti-allergic and anti-cancer effects in modern medicine [1,2]. The source material for the medicine is the root of the plant *Stellaria dichotoma* L. var. *lanceolata* Bge. (hereinafter referred to as YCH), and Ningxia, China is the original producing area of YCH. In recent years, with a lack of wild YCH sources and the successful introduction and domestication of YCH, cultivated YCH has gradually become the main source for commercial production. Changes in the mode of production can alleviate the shortage of Chinese wild herbal resources, but also change, for example, the origin, habitat and management measures of herbal medicines. The metabolites of medicinal plants are the active components of Chinese medicines that might play a therapeutic role and determine the quality of medicinal materials [3,4]. Different cultivation areas, habitats and production methods will have different effects on plant metabolites and the quality of medicinal materials [5,6]. Therefore, when adopting wild sources of medicines for cultivation, the question of whether the quality of the cultivated materials can be guaranteed needs scientific verification. At present, it is not clear what changes might have taken place in the metabolites of YCH when moving production from wild sources to cultivated plants, and whether such changes might have an impact on the quality of medicinal materials.

In this study, metabolomic technology based on ultra-high performance liquid chromatography–tandem time-of-flight mass spectrometry (UHPLC-Q-TOF MS) is used to analyze the metabolites of YCH, determine the diversity of metabolites between wild and cultivated YCH, screen for significantly different metabolites and provide reference points for the evaluation of quality in the production of YCH.

## 2. Results

### 2.1. Comparison of the Characteristics of Medicinal Materials, Contents of Methanol Extracts, Total Sterols and Total Flavonoids of Wild and Cultivated YCH

The radial texture of “Shayan” and “Zhenzhupan” and the yellow and white cross-sectional appearance are the main characteristics of YCH. As shown in Figure 1A–C, the wild and the cultivated samples I and II had the above characteristics. However, the wild samples were darker colored, brown or dark brown, while the two cultivated samples were light brown to yellow or light yellow to brown. In addition, the wild samples had generally more branches and were irregular in shape, while the cultivated samples were mostly long, column-shaped and had fewer branches. The total extractable content is the only determination index stipulated for YCH in the Chinese Pharmacopoeia, but the sterol and flavonoid contents have been reported to be important active components in YCH and are commonly used indexes to evaluate the quality. The results of content determination (Figure 1D) shows that the contents of the methanol extracts of the wild sample and the cultivated samples I and II were all higher than 20%, as stipulated in the Chinese Pharmacopoeia, and there was no significant difference among them (28%, 35% and 37%, respectively). The contents of total sterols and flavonoids of the wild sample were 4.07 g·kg^−1^ and 2.73 g·kg^−1^, respectively, values that were significantly higher than those of the cultivated YCH samples. In summary, there was no significant difference in the physical characteristics of the roots between the wild and cultivated YCH, except color and morphology. The contents of total sterols and total flavonoids in the wild YCH were significantly higher than in the cultivated YCH, while there was no significant difference in the contents of methanol extracts between the cultivated and wild YCH.

### 2.2. YCH Metabolite Detection and Quality Control Analysis

All samples were mixed thoroughly to prepare quality control (QC) samples, which were tested repeatedly 11 times. The total ion chromatograms of all QC samples were compared using spectral overlap, as shown in Appendix A. The response intensity and retention time of each chromatographic peak essentially overlapped, indicating that the precision of the instrument was good throughout the experiment. The proportion of characteristic peaks with a relative standard deviation of <30% exceeded 70% (Appendix A), indicating the good stability of the instrument. Principal component analysis (PCA) (Figure 2A,B) showed that QC samples were closely clustered together, and the samples from the same source were clustered together, indicating that the sample repeatability was good. In total, 1586 metabolites were identified by searching the local in-house standard database. Among them, 880 substances were identified in the positive (pos) ion detection mode and 706 in negative ion (neg). The 1586 substances were divided into 13 categories and more than 50 subcategories. As shown in Appendix A, the main categories were lipids and lipid-like molecules, organic acids and derivatives, organic heterocyclic compounds, phenylpropanoids and polyketides, and benzenoids. The main subcategories were carboxylic acids and derivatives, fatty acids, benzene and substituted derivatives, flavonoids and isoflavones.

### 2.3. Screening of Differential Metabolites of YCH in Different Comparable Groups

Cluster heat map analysis was performed on all the metabolites of YCH, and the results are shown in Figure 3. Samples from the same source had similar metabolite characteristics and were clustered together, while samples from different sources showed abundant differences in metabolites, indicating substantial metabolite diversity among YCH from different sources. The cluster analysis of the metabolites in cation mode detection clustered wild sample and cultivated sample I metabolites into the same group, and cultivated sample II metabolites into a separate cluster. In contrast, in anion mode identification, the analysis clustered cultivated samples I and II together, with wild sample metabolites being clustered in a separate cluster. To further analyze the differential metabolites between the wild and cultivated samples, fold-change (FC) analysis and *t*-tests were used for univariate statistical analysis. The differential metabolites were screened with FC < 0.67 or FC > 1.5, and *p* < 0.05. The results are shown in Figure 4. Compared with cultivated sample I, 392 substances among the wild sample metabolites were significantly up-regulated (224 substances in the cationic mode and 168 in anionic mode), and 339 substances were significantly down-regulated (199 substances in cationic mode and 140 in anionic mode). In addition, compared with the cultivated sample II, 211 substances among the wild sample metabolites were significantly up-regulated (161 substances in cationic mode and 50 in anionic mode), and 488 substances were significantly down-regulated (246 substances in cationic mode and 242 in anionic mode).

Orthogonal partial least squares discriminant analysis (OPLS-DA) was used for performing the multi-dimensional statistical analysis of metabolites, and an OPLS-DA model was constructed. As shown in Figure 5, the OPLS-DA model clearly distinguished the two comparison samples, and the model evaluation parameter Q^2^ was greater than 0.5, indicating that the model was stable and reliable. The variable weight value (VIP) obtained by the OPLS-DA model was set to greater than 1 as an indicator to screen differential metabolites. Using this method, 342 differential metabolites with VIP > 1 were detected in a comparison between the wild sample and cultivated sample I (191 substances in the cationic mode and 151 in the anionic mode), and 389 differential metabolites with VIP > 1 were detected comparing the wild sample and cultivated sample II (191 substances in the cationic mode and 198 in the anionic mode).

To screen the significant differential metabolites in further detail, FC analysis, *t*-tests and OPLS-DA were combined to obtain evaluation indexes, namely FC value < 0.67 or >1.5, *p* < 0.05 and VIP > 1. In this way, 230 differential metabolites (127 substances in the cationic mode and 103 in the anionic mode) were detected in a comparison between the wild sample and cultivated sample I, and 225 differential metabolites (118 substances in cationic mode and 107 in anionic mode) from the comparison between the wild sample and cultivated sample II. These differential metabolites were mainly lipids and lipid-like molecules, organic acids and their derivatives, phenylpropanoids and polyketides, and benzenoids (Figure 6).

### 2.4. Analysis of Significantly Different Metabolites between Wild and Cultivated YCH

To further screen the significant differential metabolites between wild and cultivated YCH, the differential metabolites were analyzed using a Venn diagram and screened for differential metabolites in common between the two comparison groups. As shown in Figure 7, 124 common differential metabolites were identified. All the differential metabolites in common were analyzed by cluster heat map (Figure 8). All the samples of the cultivated samples I and II were clustered into the same group, while the wild sample formed a separate cluster, which effectively distinguished the cultivated and wild samples. In addition, according to the clustering results and relative contents of 124 differential metabolites in common, 97 significantly different metabolites between the wild and cultivated samples were selected. The information of these significantly different metabolites is shown in Appendix A. The relative contents of 53 metabolites were significantly higher in the wild sample than in the cultivated samples, including 10 superclasses of substances, such as lipids and lipid-like molecules (12), organic acids and derivatives (8), phenylpropanoids and polyketides (6), alkaloids and derivatives (2), lignans, neolignans and related compounds (1), and organometallic compounds (1). These substances might be used as potential characteristic metabolites of the wild YCH. Furthermore, the relative contents of 44 metabolites were significantly higher in the cultivated samples than in the wild sample, including seven superclasses of substances, such as lipids and lipid-like molecules (19), organic acids and derivatives (3) and phenylpropanoids and polyketides (3), but not including alkaloids and derivatives, lignans, neolignans and related compounds, nor organometallic compounds. These enriched substances might be used as potential characteristic metabolites of cultivated YCH. The selected 97 significantly different metabolites were subjected to KEGG pathway enrichment analysis, and 48 metabolites among them were successfully matched with the KEGG database. The main pathways that were enriched were ascorbate and aldarate metabolism, aminoacyl-tRNA biosynthesis, histidine metabolism and beta-alanine metabolism (Figure 9).

## 3. Discussion

The Chinese Pharmacopoeia (2020 edition) requires that the methanol extract of YCH should not be less than 20.0% [2], with no other quality evaluation indicators specified. The results of this study show that the contents of the methanol extracts of the wild and cultivated samples both met the pharmacopoeia standard, and there was no significant difference among them. Therefore, there was no apparent quality difference between wild and cultivated samples, according to that index. However, the contents of total sterols and total flavonoids in the wild samples were significantly higher than those in the cultivated samples. Further metabolomic analysis revealed abundant metabolite diversity between the wild and cultivated samples. Additionally, 97 significantly different metabolites were screened out, which are listed in the Appendix A. Among these significantly different metabolites are β-sitosterol (ID is M397T42) and quercetin derivatives (M447T204_2), which have been reported to be active ingredients. Previously unreported constituents, such as trigonelline (M138T291_2), betaine (M118T277_2), fustin (M269T36), rotenone (M241T189), arctiin (M557T165) and loganic acid (M399T284_2), were also included among the differential metabolites. These components play various roles in anti-oxidation, anti-inflammatory, scavenging free radicals, anti-cancer and treating atherosclerosis and, therefore, might constitute putative novel active components in YCH. The content of active ingredients determines the efficacy and quality of the medicinal materials [7]. In summary, methanol extract as the only YCH quality evaluation index has some limitations, and more specific quality markers need to be further explored. There were significant differences in total sterols, total flavonoids and the contents of many other differential metabolites between the wild and cultivated YCH; so, there were potentially some quality differences between them. At the same time, the newly discovered potential active ingredients in YCH might have an important reference value for the study of the functional basis of YCH and the further development of YCH resources.

The importance of genuine medicinal materials has long been recognized in the specific region of origin for producing Chinese herbal medicines of excellent quality [8]. High quality is an essential attribute of genuine medicinal materials, and habitat is an important factor affecting the quality of such materials. Ever since YCH began to be used as medicine, it has long been dominated by wild YCH. Following the successful introduction and domestication of YCH in Ningxia in the 1980s, the source of Yinchaihu medicinal materials gradually shifted from wild to cultivated YCH. According to a previous investigation into YCH sources [9] and the field investigation of our research group, there are significant differences in the distribution areas of the cultivated and wild medicinal materials. The wild YCH is mainly distributed in the Ningxia Hui Autonomous Region of the Shaanxi Province, adjacent to the arid zone of Inner Mongolia and central Ningxia. In particular, the desert steppe in these areas is the most suitable habitat for YCH growth. In contrast, the cultivated YCH is mainly distributed to the south of the wild distribution area, such as Tongxin County (Cultivated I) and its surrounding areas, which has become the largest cultivation and production base in China, and Pengyang County (Cultivated II), which is located in a more southern area and is another producing area for cultivated YCH. Moreover, the habitats of the above two cultivated areas are not desert steppe. Therefore, in addition to the mode of production, there are also significant differences in the habitat of the wild and cultivated YCH. Habitat is an important factor affecting the quality of herbal medicinal materials. Different habitats will affect the formation and accumulation of secondary metabolites in the plants, thereby affecting the quality of medicinal products [10,11]. Therefore, the significant differences in the contents of total flavonoids and total sterols and the expression of the 53 metabolites that we found in this study might be the result of field management and habitat differences.

One of the main ways that the environment influences the quality of medicinal materials is by exerting stress on the source plants. Moderate environmental stress tends to stimulate the accumulation of secondary metabolites [12,13]. The growth/differentiation balance hypothesis states that, when nutrients are in sufficient supply, plants primarily grow, whereas when nutrients are deficient, plants mainly differentiate and produce more secondary metabolites [14]. Drought stress caused by water deficiency is the main environmental stress faced by plants in arid areas. In this study, the water condition of the cultivated YCH is more abundant, with annual precipitation levels significantly higher than those for the wild YCH (water supply for Cultivated I was about 2 times that of Wild; Cultivated II was about 3.5 times that of Wild). In addition, the soil in the wild environment is sandy soil, but the soil in the farmland is clay soil. Compared with clay, sandy soil has a poor water retention capacity and is more likely to aggravate drought stress. At the same time, the cultivation process was often accompanied by watering, so the degree of drought stress was low. Wild YCH grows in harsh natural arid habitats, and therefore it may suffer more serious drought stress.

Osmoregulation is an important physiological mechanism by which plants cope with drought stress, and alkaloids are important osmotic regulators in higher plants [15]. Betaines are water-soluble alkaloid quaternary ammonium compounds and can act as osmoprotectants. Drought stress can reduce the osmotic potential of cells, while osmoprotectants preserve and maintain the structure and integrity of biological macromolecules, and effectively alleviate the damage caused by drought stress to plants [16]. For example, under drought stress, the betaine content of sugar beet and *Lycium barbarum* increased significantly [17,18]. Trigonelline is a regulator of cell growth, and under drought stress, it can extend the length of the plant cell cycle, inhibit cell growth and lead to cell volume shrinkage. The relative increase in solute concentration in the cell enables the plant to achieve osmotic regulation and enhance its ability to resist drought stress [19]. JIA X [20] found that, with an increase in drought stress, *Astragalus membranaceus* (a source of traditional Chinese medicine) produced more trigonelline, which acts to regulate osmotic potential and improve the ability to resist drought stress. Flavonoids have also been shown to play an important role in plant resistance to drought stress [21,22]. A large number of studies have confirmed that moderate drought stress was conducive to the accumulation of flavonoids. Lang Duo-Yong et al. [23] compared the effects of drought stress on YCH by controlling water-holding capacity in the field. It was found that drought stress inhibited the growth of roots to a certain extent, but in moderate and severe drought stress (40% field water holding capacity), the total flavonoid content in YCH increased. Meanwhile, under drought stress, phytosterols can act to regulate cell membrane fluidity and permeability, inhibit water loss and improve stress resistance [24,25]. Therefore, the increased accumulation of total flavonoids, total sterols, betaine, trigonelline and other secondary metabolites in wild YCH might be related to high-intensity drought stress.

In this study, KEGG pathway enrichment analysis was performed on the metabolites that were found to be significantly different between the wild and cultivated YCH. The enriched metabolites included those involved in the pathways of ascorbate and aldarate metabolism, aminoacyl-tRNA biosynthesis, histidine metabolism and beta-alanine metabolism. These metabolic pathways are closely related to plant stress resistance mechanisms. Among them, ascorbate metabolism plays an important role in plant antioxidant production, carbon and nitrogen metabolism, stress resistance and other physiological functions [26]; aminoacyl-tRNA biosynthesis is an important pathway for protein formation [27,28], which is involved in the synthesis of stress-resistant proteins. Both histidine and β-alanine pathways can enhance plant tolerance to environmental stress [29,30]. This further indicates that the differences in metabolites between the wild and cultivated YCH was closely related to the processes of stress resistance.

Soil is the material basis for the growth and development of medicinal plants. Nitrogen (N), phosphorus (P) and potassium (K) in soil are important nutrient elements for the growth and development of plants. Soil organic matter also contains N, P, K, Zn, Ca, Mg and other macroelements and trace elements required for medicinal plants. Excessive or deficient nutrients, or unbalanced nutrient ratios, will affect the growth and development and the quality of medicinal materials, and different plants have different nutrient requirements [31,32,33]. For example, a low N stress promoted the synthesis of alkaloids in Isatis indigotica, and was beneficial to the accumulation of flavonoids in plants such as Tetrastigma hemsleyanum, Crataegus pinnatifida Bunge and Dichondra repens Forst. In contrast, too much N inhibited the accumulation of flavonoids in species such as Erigeron breviscapus, Abrus cantoniensis and Ginkgo biloba, and affected the quality of medicinal materials [34]. The application of P fertilizer was effective in increasing the content of glycyrrhizic acid and dihydroacetone in Ural licorice [35]. When the application amount exceeded 0·12 kg·m^−2^, the total flavonoid content in Tussilago farfara decreased [36]. The application of a P fertilizer had a negative effect on the content of polysaccharides in the traditional Chinese medicine rhizoma polygonati [37], but a K fertilizer was effective in increasing its content of saponins [38]. Applying 450 kg·hm^−2^ K fertilizer was the best for the growth and saponin accumulation of two-year-old Panax notoginseng [39]. Under the ratio of N:P:K = 2:2:1, the total amounts of hydrothermal extract, harpagide and harpagoside were the highest [40]. The high ratio of N, P and K was beneficial to promote the growth of Pogostemon cablin and increase the content of volatile oil. A low ratio of N, P and K increased the content of the main effective components of Pogostemon cablin stem leaf oil [41]. YCH is a barren-soil-tolerant plant, and it might have specific requirements for nutrients such as N, P and K. In this study, compared with the cultivated YCH, the soil of the wild YCH plants was relatively barren: the soil contents of organic matter, total N, total P and total K were about 1/10, 1/2, 1/3 and 1/3 those of the cultivated plants, respectively. Therefore, the differences in soil nutrients might be another reason for the differences between the metabolites detected in the cultivated and wild YCH. Weibao Ma et al. [42] found that the application of a certain amount of N fertilizer and P fertilizer significantly improved the yield and quality of seeds. However, the effect of nutrient elements on the quality of YCH is not clear, and fertilization measures to improve the quality of medicinal materials need further study.

Chinese herbal medicines have the characteristics of “Favorable habitats promote yield, and unfavorable habitats improve quality” [43]. In the process of a gradual shift from wild to cultivated YCH, the habitat of the plants changed from the arid and barren desert steppe to fertile farmland with more abundant water. The habitat of the cultivated YCH is superior and the yield is higher, which is helpful to meet the market demand. However, this superior habitat led to significant changes in the metabolites of YCH; whether this is conducive to improving the quality of YCH and how to achieve a high-quality production of YCH through science-based cultivation measures will require further research.

Simulative habitat cultivation is a method of simulating the habitat and environmental conditions of wild medicinal plants, based on knowledge of the long-term adaptation of the plants to specific environmental stresses [43]. By simulating various environmental factors that affect the wild plants, especially the original habitat of plants used as sources of authentic medicinal materials, the approach uses scientific design and innovative human intervention to balance the growth and secondary metabolism of Chinese medicinal plants [43]. The methods aims to achieve the optimal arrangements for the development of high-quality medicinal materials. Simulative habitat cultivation should provide an effective way for the high-quality production of YCH even when the pharmacodynamic basis, quality markers and response mechanisms to environmental factors are unclear. Accordingly, we suggest that scientific design and field management measures in the cultivation and production of YCH should be carried out with reference to the environmental characteristics of wild YCH, such as arid, barren and sandy soil conditions. At the same time, it is also hoped that researchers will conduct more in-depth research on the functional material basis and quality markers of YCH. These studies can provide more effective evaluation criteria for YCH, and promote the high-quality production and sustainable development of the industry. 

## 4. Materials and Methods

### 4.1. Experimental Materials

The experimental materials in this study were the roots of the *Stellaria dichotoma* L.var. *lanceolata* Bge., which were collected in August 2020. Among the samples, the wild-type samples (Wild) were collected from Lingwu City and Yinchuan City, Ningxia Hui Autonomous Region, and the cultivated samples were collected from Tongxin County, Wuzhong City, Ningxia Hui Autonomous Region (Cultivated I) and Pengyang County, Guyuan City, Ningxia Hui Autonomous Region (Cultivated II). Additionally, they had more than 3 years of growth, and information on the place of collection is shown in Table 1. Each sampling spot was randomly sampled, and there were 6 biological reduplicative samples. These samples were naturally dried to a constant weight and crushed, passed through a 40-mesh sieve, and stored frozen in a dark place for later use.

### 4.2. Characteristics of the Medicinal Materials and the Content Detection of Extract, Total Flavonoids and Total Sterols

Characteristics of the medicinal materials: With reference to the ‘Pharmacopoeia of the People’s Republic of China (Volume I)’ (2020 Edition) under the ‘Stellariae Radix’ provisions’, the characteristics of the medicinal materials, the sample ‘Shayan’, ‘Zhenzhupan’, with a yellow and white cross-sectional appearance, were observed and compared.

Extract content: With reference to the ‘Pharmacopoeia of the People’s Republic of China (Volume IV)’ (2020 Edition) in the ‘2201 extract detection method’, we used an alcohol-soluble extract detection method for the detection of the methanol extract content.

Total flavonoid content: According to the methods in the literature [44], 2.00 g of medicinal powder sample was accurately weighed into a centrifuge tube, and 25 mL of 95% ethanol was added for ultrasonic extraction for 30 min. The supernatant was separated, and the residue was added with 25 mL of 95% ethanol for ultrasonic extraction for 15 min. Then, the supernatant of the two extractions was mixed as the tested sample solution, and rutin was used as the reference substance; the absorbance of the sample was measured at a wavelength of 496 nm. The total flavonoids content of the medicinal material was thus determined.

Total sterol content: According to the literature [44], 0.50 g medicinal powder was accurately weighed and placed in a 25 mL volumetric flask, added with 20 mL chloroform, and then ultrasonically extracted for 20 min. Then, it was cooled and diluted with chloroform to the scale, and shaken and filtered to obtain the tested solution. The absorbance was measured at a wavelength of 546 nm with α-spinasterol as the reference substance, and the total sterol content of the medicinal material was calculated.

### 4.3. Metabolic Substance Detection

Metabolomics analysis was adopted from a previous method as we reported elsewhere [44]. The separation was performed with Agilent 1290 Infinity LC HILIC column; column temperature of 25 °C; flow rate of 0.5 mL/min; injection volume of 2 μL; mobile phase composition A: water + 25 mMol ammonium acetate + 25 mMol ammonia, and B: acetonitrile; gradient elution procedure as follows: 0~0.5 min, 95% B; 0.5~7 min, B linearly varied from 95% to 65%; 7~8 min, B linearly varied from 65% to 40%; 8~9 min, B maintained at 40%; 9~9.1 min, B linearly varied from 40% to 95%; 9.1~12 min, B maintained at 95% [44]. Mass spectrometric analysis was performed with a triple TOF 6600 mass spectrometer, and the positive and negative ionF modes of electrospray spray ionization (ESI) were used for detection. The metabolites were identified by means of searching the local self-built standard database established by Shanghai Applied Protein Technology [44].

### 4.4. Data Analysis

Principal component analysis (PCA), fold change analysis (FC), *t*-test, orthogonal partial least squares discriminant analysis (OPLS-DA), cluster heatmap analysis, volcano chart analysis, Wayne chart analysis and KEGG pathway enrichment analysis were performed on the relevant data by using statistical analysis software, such as Excel, SPSS Statistics 22, Origin 2018 and R software (3.6.1), and online analysis platforms, such as Zhongke New Life Bioinformatics Cloud Platform and Metabo Analyst 5.0.

## 5. Conclusions

In this study, the content of the methanol extracts of both the wild and cultivated YCH exceeded the standard content specified by the Chinese Pharmacopoeia. However, the contents of total sterols and total flavonoids were significantly higher in the wild YCH than in the cultivated YCH. Further metabolomic analysis showed additional significant differences in metabolites between the wild and cultivated YCH. We detected 97 significantly different metabolites between the wild and cultivated YCH, including β-sitosterol and quercetin derivatives, as well as many newly-discovered potential active components, such as trigonelline, arctiin and loganic acid. These significantly different metabolites were enriched in metabolic signaling pathways, such as ascorbic acid and aldehyde acid metabolism, aminoacyl-tRNA biosynthesis, histidine metabolism and β-alanine metabolism. Our results indicate that, in changing the source of YCH from wild to cultivated plants, not only the production mode changed, but also the quality and amounts of metabolites in YCH were significantly affected. This research provides valuable knowledge as a basis for YCH quality evaluation and high-quality production.

## Figures and Tables

**Figure 1 plants-12-00775-f001:**
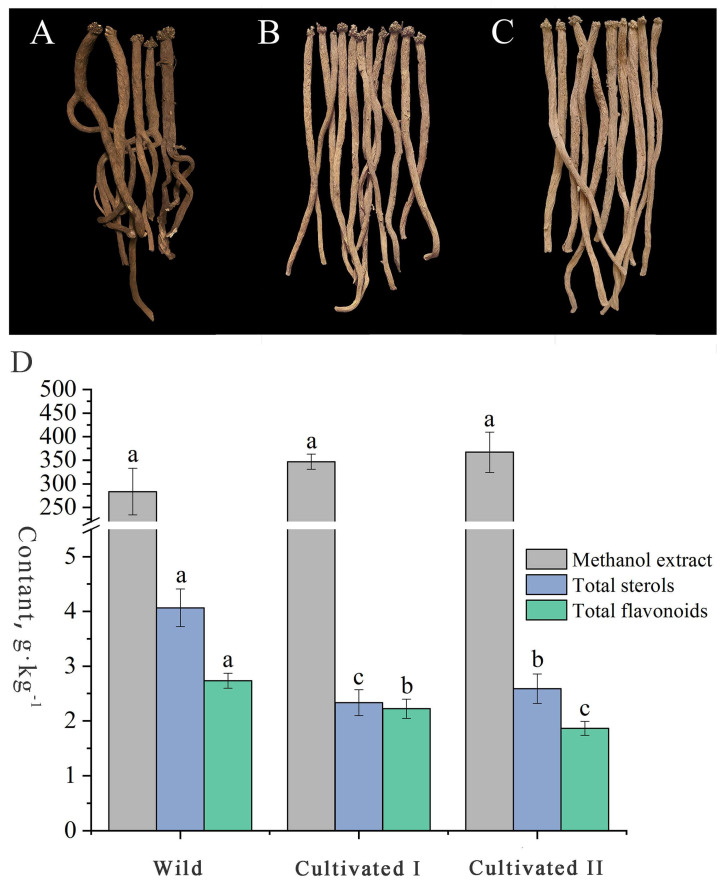
Characteristics and metabolite contents of the wild and cultivated YCH. (**A**–**C**) Overall characteristics of Wild (wild), Cultivated I and Cultivated II (cultivated), roots, respectively. (**D**)The content of methanol extract, total sterols and total flavonoids of Wild, Cultivated I and Cultivated II, the lowercase letters in the figure indicate the significant difference between the samples (*p* < 0.05), the same letter represents no significant difference, and different letters represent significant difference.

**Figure 2 plants-12-00775-f002:**
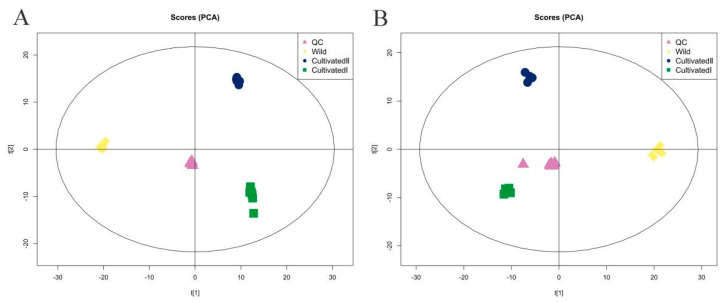
PCA score plot of YCH from different sources. (**A**) Pos mode; (**B**) neg mode.

**Figure 3 plants-12-00775-f003:**
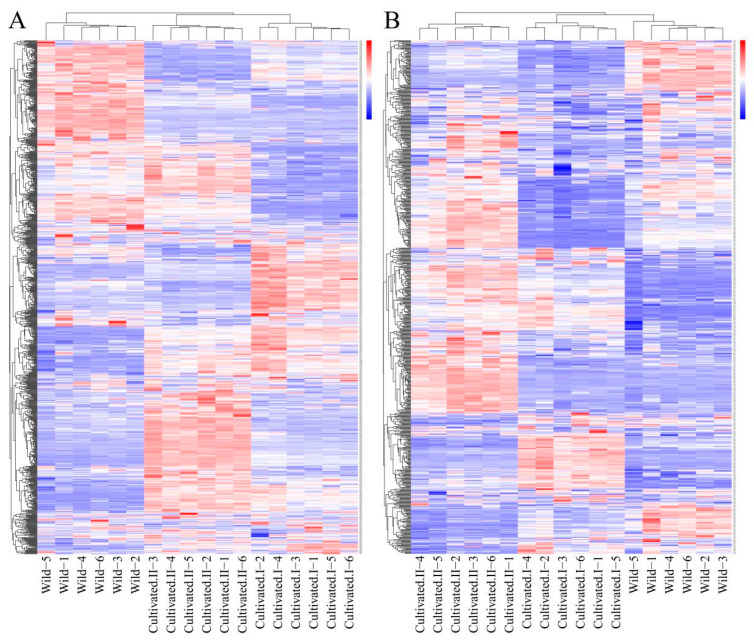
Cluster heatmap analysis of metabolites from different sources of YCH. (**A**) The metabolites detected in the positive ion mode. (**B**) The metabolites detected in the negative ion mode.

**Figure 4 plants-12-00775-f004:**
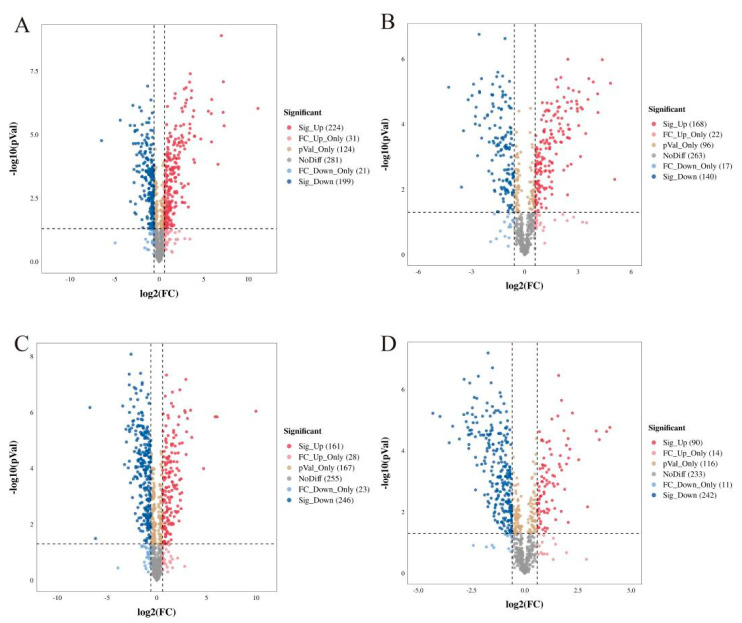
Volcano diagram of differential metabolites from different sources of YCH. (**A**,**B**) The differential metabolites of Wild and Cultivated I detected in the positive and negative ion modes, respectively. (**C**,**D**) The differential metabolites of Wild and Cultivated II detected in the positive and negative ion modes, respectively. The plots are of log probability vs. fold change, with color coding for different categories of significant differences.

**Figure 5 plants-12-00775-f005:**
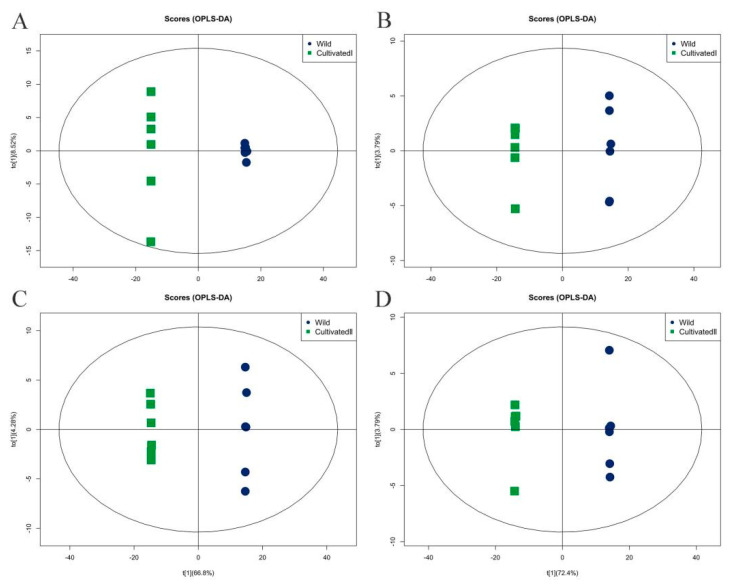
Orthogonal partial least squares discriminant analysis of different origins of YCH. (**A**,**B**) The score plots of Wild and Cultivated I in the positive and negative ion modes, respectively, with Q^2^ of 0.997 and 0.998, respectively. (**C**,**D**) The score plots of Wild and Cultivated II in the cation and anion modes, respectively, with Q^2^ of 0.997 and 0.998, respectively.

**Figure 6 plants-12-00775-f006:**
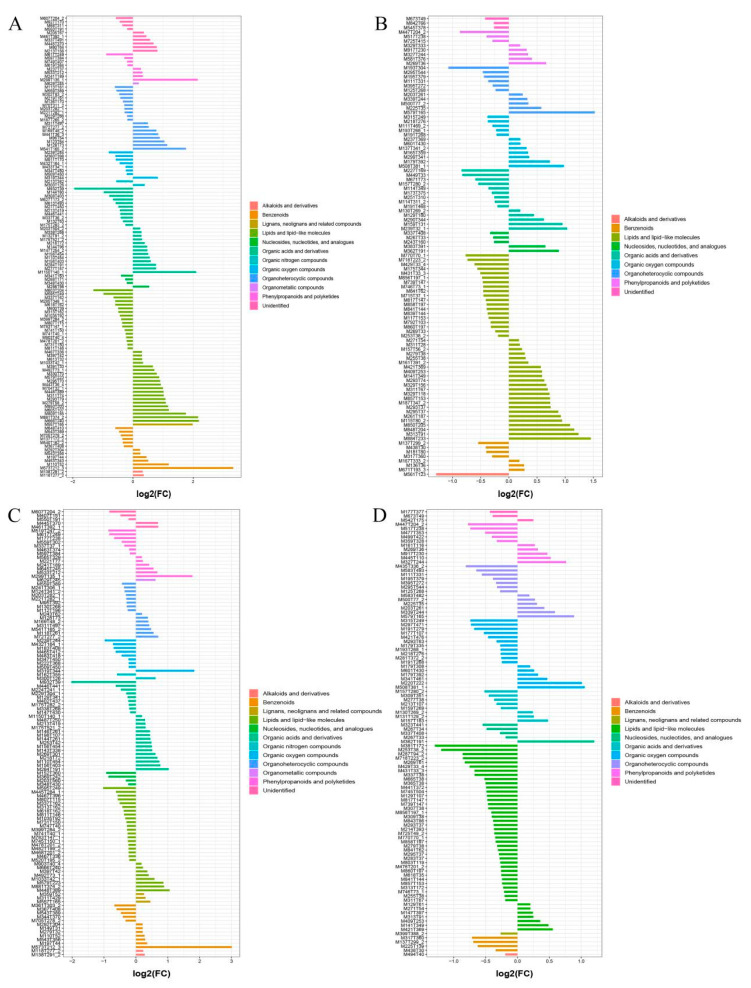
Classes of significantly different metabolites in different sources of YCH. (**A**,**B**) The fold-change histograms of significantly different metabolites between Wild and Cultivated I in the positive and negative ion mode, respectively. (**C**,**D**) The fold-change histograms of significantly different metabolites between Wild and Cultivated II in the cation and anion modes, respectively.

**Figure 7 plants-12-00775-f007:**
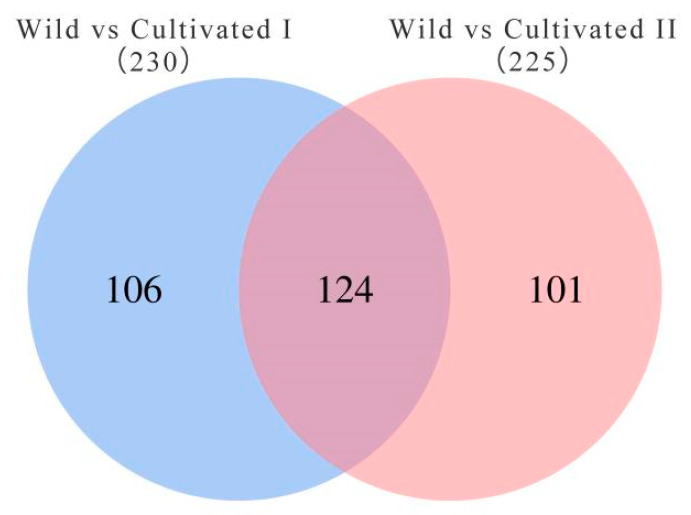
Venn diagram of the number of metabolites with significant differences between the different sources of YCH.

**Figure 8 plants-12-00775-f008:**
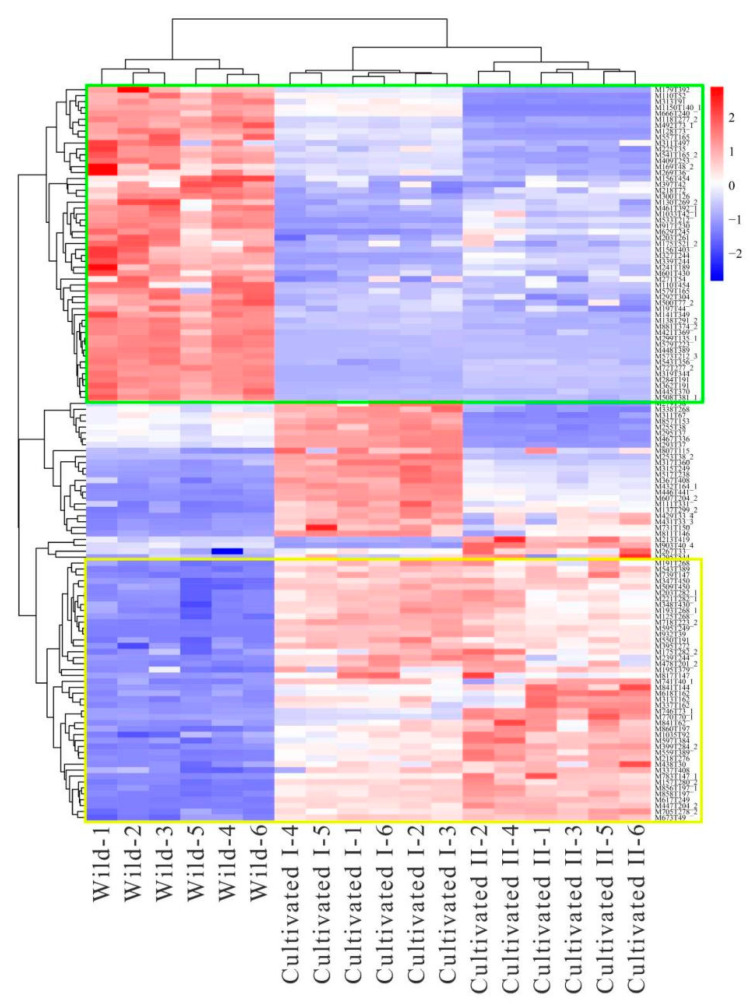
Cluster heat map of significantly different metabolites from different sources of YCH. Metabolites marked in the green box are potential characteristic metabolites of Wild; metabolites marked in the yellow box are potential characteristic metabolites of Cultivated I and Cultivated II.

**Figure 9 plants-12-00775-f009:**
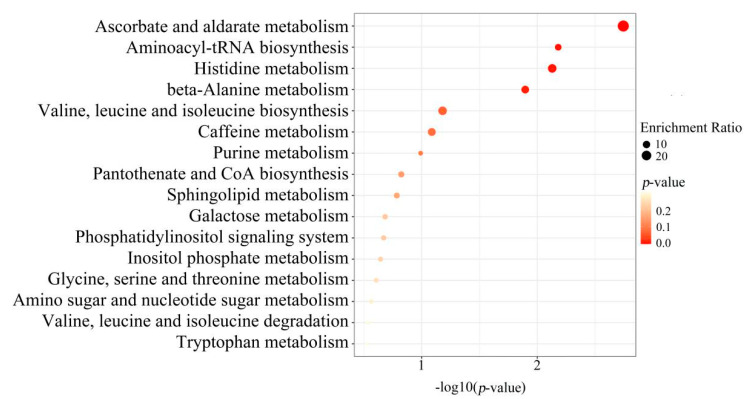
KEGG pathway analysis: enrichment of metabolites with a significant difference between Wild, Cultivated I and Cultivated II.

**Table 1 plants-12-00775-t001:** Information of the collection place.

Label	Latitude, N	Longitude, E	Elevation, m	Average Annual Precipitation, mm	Average Annual Temperature, °C	Soil Texture	pH Value	Organic Matter Content, g·kg^−1^	Total Nitrogen Content g·kg^−1^	Total Phosphorus Content g·kg^−1^	Total Potassium Content, g·kg^−1^
Wild	38.05°	106.59°	1273	304.40	9.22	sandy soil	9.17	0.72	0.28	0.19	1.00
Cultivated I	36.76°	106.36°	1558	311.15	8.78	clayey soil	8.71	7.00	0.45	0.55	3.32
Cultivated II	35.75°	106.80°	1395	583.17	8.90	clayey soil	8.51	8.03	0.56	0.60	3.23

## Data Availability

Data is contained within the article or Appendix A.

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
