# Peer review of "Metabolomic Analysis Reveals the Metabolic Diversity of Wild and Cultivated Stellaria Radix (Stellaria dichotoma L. var. lanceolata Bge.)"

_plants, 2023, doi:10.3390/plants12040775_

Round 1

Reviewer 1 Report

I have carefully examined the manuscript entitled “Metabonomics Reveals Metabolic Diversity of Wild and Cultivated Stellaria Radix (Stellaria dichotoma L. var. lanceolata Bge.)” by Li et al. Authors describe an efficient approach to measure changes in the metabolome between cultivated and wild Yinchaihu (YCH), a traditional Chinese medicine coming from the roots of the plant Stellaria dichotoma.

Ciìonyinuing their previous studies on the analysis of Stellaria dichotoma content and its biological and pharmacological potential application, authors put their efforts to start a comparative MS-based metabolomic analysis of YCH coming from the cultivated or wild plant of origin, in order to give evidence of potential quantitative differences in the metabolome expression.

They compared metabolome levels of cultivated and wild samples measuring the content of methanol extracts and the content of the total sterols and flavonoids, that they selected as relevant markers in this analysis.

Authors performed a correct work, collecting a huge amount of data that were submitted to a very accurate quality control analysis. Through application of diverse statistical approaches, authors detected the most significant difference between wild and cultivated YCH in the sterols and flavonoids content. Although they didn’t show any experimental correlations between these metabolic differences and the biological activity of the two medicinal preparations, I think that this study has a relevance as starting point for following investigations.

The results of their comprehensive metabolomic approach is impressive and of high quality.

On this basis, I consider this work interesting and significant in the field of traditional chinese medicine. Moreover, the manuscript needs limited language revision.

My only comments are:

1. I don’t think that the expression “Metabonomic” could apply to this kind of analysis, and “Metabolomic” should be more appropriate.

2. I found the discussion chapter a little confused in the parts starting from row 225, where authors described the potential influence of habitats and environmental stress, then osmoregulation and alkaloids level as a response to drought stress, and the importance of nitrogen, phosphorus and potassium levels in different soils. In this part of the discussion some relevant points occurred, but I would ask the authors to rephrase the discussion (maybe some comments on other cites papers are too much extended) for more clarity.

On this basis, and after minor revisions, I believe that the above-mentioned manuscript will be suitable for publication on Plants

Author Response

Dear reviewer:

Thanks very much for taking the time to review this manuscript. We really appreciate all your professional comments and suggestions. Those comments are all valuable and very helpful for revising and improving our paper, as well as the important guiding significance to our research. We have studied the comments carefully and have made corrections which we hope meet with approval. Revised portions are marked in red in the manuscript. Please find our revisions in the re-submitted files. The main corrections in the paper and the responses to your comments are as follows:

â‘  I don’t think that the expression “Metabonomic” could apply to this kind of analysis, and “Metabolomic” should be more appropriate.

Reply: Thank you for your valuable suggestions. We strongly agree with your views and have revised them(Lines 2 and 50). Please check, thank you.

â‘¡I found the discussion chapter a little confused in the parts starting from row 225, where authors described the potential influence of habitats and environmental stress, then osmoregulation and alkaloids level as a response to drought stress, and the importance of nitrogen, phosphorus and potassium levels in different soils. In this part of the discussion some relevant points occurred, but I would ask the authors to rephrase the discussion (maybe some comments on other cites papers are too much extended) for more clarity.

Reply:Thank you for pointing out the problem. In response to your questions, we re-examined the ' discussion ' section and made adjustments and additions. In particular, the logical relationship of the discussion and the accuracy of the statement were comprehensively modified. I hope this modification can solve the existing problems. Please consult, thank you. Stellaria dichotoma L. var. lanceolata Bge. (YCH) is a drought-tolerant and barren-tolerant plant. We found that when YCH changed from wild to cultivated, its metabolites changed significantly, and its habitat changed significantly, especially water and soil nutrients. Therefore, in the ' discussion ' section, we focused on these two points for discussion and outlook. In the following research work, we will also focus on these two issues for further research.

â‘¢The manuscript needs limited language revision.

Reply: Thank you for your valuable advice. We asked the doctor whose mother tongue is English to re-examine the language problems of the full text and make modifications. The proof of polishing is shown in the annex. Thank you.

We hope that the revision is acceptable, and we look forward to hearing from you soon.

Best wishes!

Your sincerely,

Li Peng

Zhenkai Li

Reviewer 2 Report

The authors have made a good effort to show the difference of metabolites between plants using different statistical methods.

1- Considering that the methanolic extract has limitations, why were different fractions or at least aqueous-alcoholic extracts not used to compare different materials?

2- Different seasons can have a distinctive effect on the amount of effective metabolites. This factor is not considered in methods and statistical analysis.

Author Response

Dear reviewer:

Thanks very much for taking the time to review this manuscript. We really appreciate all your professional comments and suggestions. Those comments are all valuable and very helpful for revising and improving our paper, as well as the important guiding significance to our research. We have studied the comments carefully and have made corrections which we hope meet with approval. Revised portions are marked in red in the manuscript. Please find our revisions in the re-submitted files. The main corrections in the paper and the responses to your comments are as follows:

①Considering that the methanolic extract has limitations, why were different fractions or at least aqueous-alcoholic extracts not used to compare different materials?

Reply: Thank you for your valuable suggestions, which is very helpful for our next research work. At present, methanol extract is the only content determination index of YCH ï¼ˆStellaria dichotoma L. var. lanceolata Bge.) in Chinese Pharmacopoeia. In addition, there is no clear quality marker. This also brings great difficulties to YCH quality evaluation. Therefore, we used metabolomics techniques to find the differences in the composition of wild and cultivated YCH as much as possible. Whether there are differences in different fractions or different solvent extracts is a good research idea. Thank you again for your suggestions. We will carry out relevant research work in the future research work and strive to achieve more valuable research results.

â‘¡Different seasons can have a distinctive effect on the amount of effective metabolites. This factor is not considered in methods and statistical analysis.

Reply: Thank you for your valuable suggestions. Your questions and suggestions are very critical. Season and harvest time are important factors affecting the accumulation of metabolites. On the one hand, in order to avoid the influence of other factors on the sample, we chose to harvest wild and cultivated YCH at the same time. In order to minimize the interference of other factors on the results of this study. In this revision, we further emphasized that all samples were collected at the same time in the ' Experimental Material ' section ï¼ˆLine 365). Please check. On the other hand, we have also paid attention to this problem. Relevant research work has been carried out and some research results have been obtained. We will show them in the next article. Thank you again for your advice, hope our reply can answer your questions.We hope that the revision is acceptable, and we look forward to hearing from you soon.

Best wishes!

Your sincerely,

Li Peng

Zhenkai Li

Round 2

Reviewer 2 Report

The revised version of manuscript is acceptable. 

Author Response

Thank you for your review and valuable suggestions, thank you for your recognition of this article, thank you.